# Robotic scrub nurse to anticipate surgical instruments based on real-time laparoscopic video analysis
Lars Wagner [1,6] ✉, Sara Jourdan [2,6], Leon Mayer[1], Carolin Müller [1], Lukas Bernhard [1], Sven Kolb [1], Farid Harb [1], Alissa Jell [1,3], Maximilian Berlet [1,3], Hubertus Feussner[1,3], Peter Buxmann[2,7], Alois Knoll[4,7] & Dirk Wilhelm[1,3,5,7]

## Abstract

**Background** Machine learning and robotics technologies are increasingly being used in the healthcare domain to improve the quality and efficiency of surgeries and to address challenges such as staff shortages. Robotic scrub nurses in particular offer great potential to address staff shortages by assuming nursing tasks such as the handover of surgical instruments.

**Methods** We introduce a robotic scrub nurse system designed to enhance the quality of surgeries and efficiency of surgical workflows by predicting and delivering the required surgical instruments based on real-time laparoscopic video analysis. We propose a three-stage deep learning architecture consisting of a single frame-, temporal multi frame-, and informed model to anticipate surgical instruments. The anticipation model was trained on a total of 62 laparoscopic cholecystectomies.

**Results** Here, we show that our prediction system can accurately anticipate 71.54% of the surgical instruments required during laparoscopic cholecystectomies in advance, facilitating a smoother surgical workflow and reducing the need for verbal communication. As the instruments in the left working trocar are changed less frequently and according to a standardized procedure, the prediction system works particularly well for this trocar.

**Conclusions** The robotic scrub nurse thus acts as a mind reader and helps to mitigate staff shortages by taking over a great share of the workload during surgeries while additionally enabling an enhanced process standardization.

## Plain language summary

Staff shortages in healthcare are an emerging problem leading to undersupply of medical experts such as scrub nurses in the operating room. The absence of these scrub nurses, who are responsible for providing surgical instruments, means that surgeries must be postponed or canceled. Robotic technologies and artificial intelligence offer great potential to address staff shortages in the operating room. We developed a robotic scrub nurse system that is able to take over the tasks of a human scrub nurse by delivering the required surgical tools. To maintain the pace of the surgery, our robotic scrub nurse system is also capable of predicting these required surgical tools in advance using artificial intelligence. The system is tested on laparoscopic cholecystectomies, a surgery, where the gallbladder is removed in a minimally invasive technique. We show that our prediction system can predict the majority of surgical instruments for this specific surgery facilitating a smoother surgical workflow and reducing the need for verbal communication. With further development, our system may help to cover the need for surgery while streamlining the surgical process through predictive support, potentially improving patient outcomes.

The convergence of big data and improved computing power has led to rapid advances in the fields of machine learning (ML) and robotics in recent years, enabling their increasing deployment in high-risk environments[1–3]. A notable example is the market for surgical robotics, which is projected to experience robust growth and reach a substantial value of \$16.77 billion by the year 2031[4]. In areas such as surgery, where precision and expertise are critical, these systems have emerged as valuable tools, capable of assisting and augmenting human experts[5,6]. Overall, the integration of robotics and

---

ML technologies in the operating room (OR) aims to achieve two primary goals: To enhance operative performance and patient outcomes, thereby improving the quality of surgeries, and to enhance the efficiency of surgical workflows[7,8]. A seamless surgical workflow relies heavily on the scrub nurse, who is responsible for passing surgical instruments and sterile goods to the surgeon in the intraoperative part of a surgery[9]. Experienced scrub nurses can anticipate the surgeon's needs without verbal communication, helping to keep concentration levels high[10,11] as well as to avoid cross-infections[11] and misunderstandings[12,13]. Especially misunderstandings regarding equipment such as surgical instruments cause delays, inefficiencies or tension within the team[14]. Therefore, both prior to and throughout the process of exchanging instruments, nonverbal communication between the surgeon and scrub nurse is aspired, which requires the scrub nurse to anticipate instruments in a timely manner[15].

However, due to the ongoing shortage of surgical personnel[16,17], experienced scrub nurses are becoming increasingly scarce, causing a profound impact on the surgical workflow. Novice scrub nurses require years of training to obtain sufficient technical and nontechnical skills (NTS). They show the poorer ability to anticipate surgeons' needs, take longer to hand over instruments, and make significantly more errors than experienced scrub nurses[18]. In general, lower NTS, which also include communication[11], affect the workflow of a surgery in terms of quality and efficiency[19–21].

To acquire contemporary insights, we carried out a pre-study survey with 50 surgeons from clinics across Germany (survey demographics and reliability metrics can be found in Supplementary Tables 1 and 2). This survey was aimed at understanding the impact of human factors, NTS-related challenges, and infrastructural challenges on the quality and efficiency of surgical workflows. In addition to the impact of emotional tensions within the operating room team, the survey identified infrastructural challenges as the primary factors adversely influencing the efficiency and quality of surgical workflows, namely staff shortages, lack of prior knowledge, and lack of process standardization. Furthermore, the frequency of occurrence of the aforementioned three infrastructural challenges was rated as being notably high. Figure 1 additionally illustrates the surgeons' assessment of how the most significant infrastructural challenges detrimentally affect the quality and efficiency of surgical workflows. For example, efficiency is particularly affected by challenges such as staff shortages. Further results of the survey can be found in Supplementary Figs. 1–10. The survey highlights the urgent need to address infrastructural challenges and derive solution approaches to overcome staffing shortages, improve the level of experience in surgical teams, and enhance process standardization.

Motivated by the surgeons' assessment in the survey, we see great value in the integration and utilization of ML and robotic technologies, such as a Robotic Scrub Nurse (RSN), which is able to predict and manage most of the handover processes of the required surgical instruments to counteract these challenges.

Research indicates that RSNs hold significant potential to mitigate staff shortages[22–28] which could improve the efficiency of surgical procedures. Moreover, high-performing RSNs offer substantial prospects for enhancing the adherence to standardized processes, while also contributing additional expertise to the surgical team. However, existing RSN approaches have so far primarily relied on voice input from the surgeon and can only deliver instruments on the surgeon's verbal request[24]. To enable such robotic systems to proactively perform simple manipulation tasks of surgical instruments, it is essential to incorporate ML algorithms to allow the RSNs to perceive and understand the surgical environment, make autonomous decisions, and adapt to changing circumstances[29]. The incorporation of ML algorithms to anticipate the surgeon's needs ensures that instruments are prepared in advance, since the delivery speed of RSNs is lower in comparison to human scrub nurses[30]. Thus, early anticipation of instrument changes reduces downtime and enables a seamless surgical workflow in terms of efficiency[31].

Previous work attempting to predict instruments prior to the surgeon's verbal request focuses primarily on recognizing gestures[32] or making rough estimates regarding the occurrence and sequence of surgical instruments[33–35]. However, these approaches do not enable RSNs to act as mind readers[36] and autonomously anticipate and prepare the required instruments in advance without the need for gestures or voice commands by the surgeon. Expanding these approaches to include time-specific anticipation of the correct instruments prior to the instrument request is a crucial task for robot-assisted surgeries in the future. Therefore, in this work we address the problems in desired nonverbal communication and seamless instrument handover described above by implementing an ML-based instrument prediction system in the RSN developed at the University Hospital rechts der Isar of the Technical University of Munich (Fig. 2). The implementation of the instrument prediction system allows the RSN to make autonomous decisions about instrument handovers and thus operate as an experienced nurse or mind reader.

We recorded over 30 hours of laparoscopic surgical procedures to train the instrument prediction system. In these video recordings, laparoscopic instruments were manually annotated by a team of experienced surgeons and medical specialists. This rigorous approach empowers the ML-based

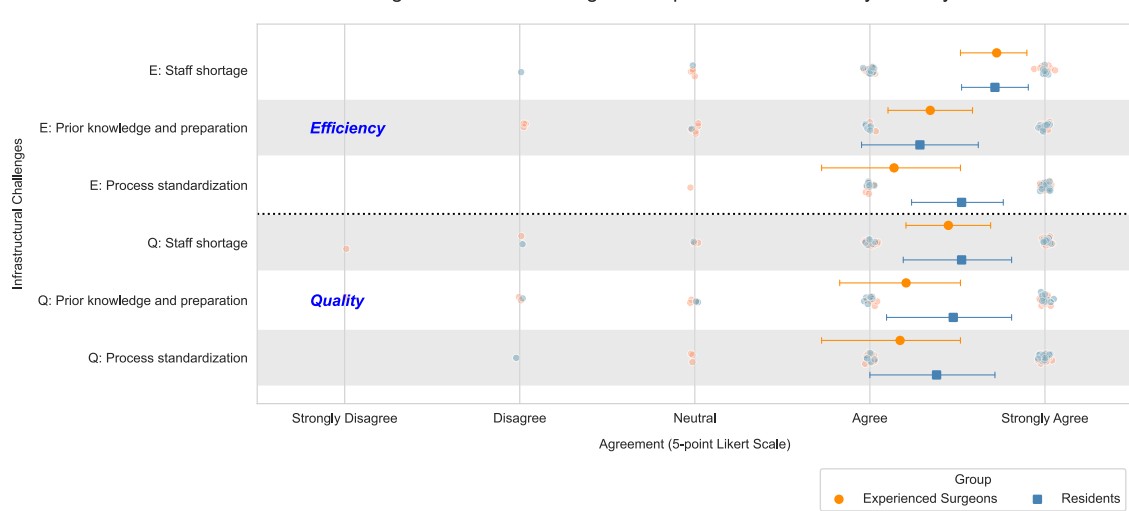

**Fig. 1 | An evaluation by surgeons on the impact of infrastructural challenges in clinics on the efficiency and quality of OR workflows.** While surgeons believe that infrastructural challenges have a negative impact on quality and efficiency, efficiency in particular is severely impaired by a shortage of staff; $n_{experienced}$ = 29; $n_{novice}$ = 21; error plots outlining mean value and respective standard deviation.

system to execute instrument anticipation tasks, enabling the RSN to proactively hand over the required surgical instruments prior to being needed in a procedure. To further bolster the system's predictive reliability, we integrate surgical domain knowledge into the model's framework, thereby enriching its anticipatory accuracy with contextual understanding. The study further presents an evaluation metric that incorporates the requirements from surgical procedures into the performance evaluation of ML and robotic systems, which reflects the time criticality of decisions made in the OR.

This study introduces a robotic scrub nurse system designed to enhance the quality of surgical procedures and the efficiency of surgical workflows. The system predicts and delivers the required surgical instruments based on real-time laparoscopic video analysis. A three-stage deep learning architecture, comprising a single-frame-, temporal multi frame-, and informed model, anticipates surgical instruments. The evaluation demonstrates that the prediction system can anticipate the majority of instrument changes, facilitating a smoother surgical workflow and reducing the need for verbal communication. The system's primary benefit is its capacity to alleviate staffing shortages by assuming a significant portion of the workload during surgical procedures, while simultaneously enhancing process standardization.

## Methods
### Prerequisites
Contextual execution of robotic autonomous actions using ML methods requires data along a surgical intervention[37]. The data can be obtained from

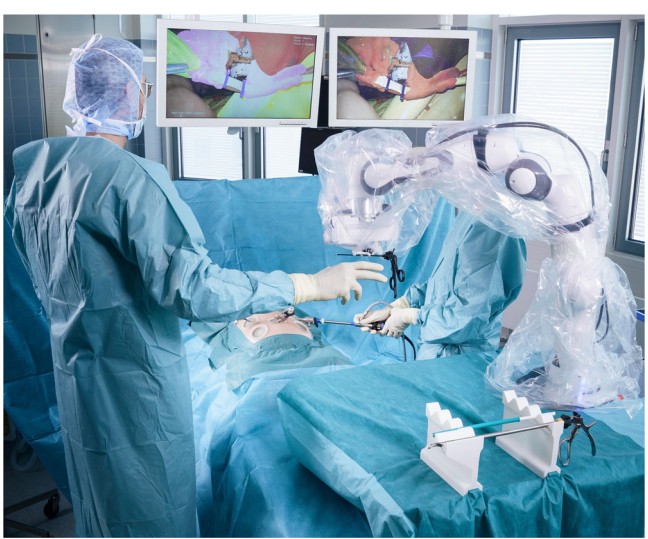

**Fig. 2 | Deployment of the developed RSN in a simulated surgery at the University Hospital rechts der Isar of the Technical University of Munich.** The RSN predicts an instrument change and passes the next required instrument to the surgeon without verbal communication. This allows the surgeon to focus on the right endoscopic screen while real-time AI assistance is provided on the left endoscopic screen.

different sensory modalities, such as vision, audio, physiological or device signals. During laparoscopic procedures, endoscopic video data is the crucial data source as it provides a view of the surgical field. Current research focuses mainly on identifying surgical phases[38–41] and detecting or segmenting surgical tools[42–47] based on laparoscopic videos[48]. Various data sets[38,44,49,50] are available for this purpose, ranging from binary annotations to segmentation masks.

In our setup, we also use the laparoscopic video signal as input for our RSN system, providing the basis for predicting the next required instrument. The RSN is located in the sterile area next to the operating table as shown in Fig. 2 and performs the tasks of a human scrub nurse. The surgeon has access to the patient's intra-abdominal area through a left and right working trocar. Thus, the surgeon inserts the laparoscopic instruments into the abdominal cavity, where they are visible in the endoscopic video. The first assistant is usually responsible for guiding the endoscope.

### Task formulation
Since unimodal data sets only contain the instrument presence within laparoscopic videos, we must calculate the robotic prediction time of the next required instrument retroactively from the time $t_0$ another instrument is last used before it is changed. We set the prediction time $t_{pred} = t_0 - \delta$ so that the prediction must take place at least $\delta$ seconds before the last instrument is extracted to ensure that the robotic assistance has enough time to prepare the next required instrument. With a certain acceptable threshold, we can identify a prediction window $w_p$ ($t_{pred} \pm \tau$), in which the instrument change should be correctly predicted. In addition, the prediction window is shifted ($\sigma$) to overlap the point of instrument extraction, allowing the model to learn on disappearance of the instrument from the endoscopic image. Considering the substantial delay the robotic arm introduces due to its slow movement pattern, a resting window $w_r$ must be established, punishing premature movement which would prevent or at least delay the correct movement in the prediction window. We also introduce a disregard window $w_d$, representing periods between prediction and resting windows in which no prediction should take place.

In contrast to previous attempts at surgical instrument anticipation[33–35], we formulate the instrument anticipation task as a classification problem. Given a frame $i$ from a video at time $t_i$, we extract the visual feature vector $V$, the detection feature vector $D$, and the segmentation feature vector $S$. Based on this observed sequence $\{V_i, D_i, S_i\}$, our model attempts to classify the anticipative status $y_\lambda(i, \alpha)$ of the next required instrument $\lambda$ depending on the surgery phase $\alpha$ in the prediction window, which triggers the robotic handover of the predicted instrument to the surgeon. During the resting window, the model should set the predictive status to idle $y_0(i, \alpha)$ to prevent the robot from making any movement. This process is illustrated in Fig. 3. Our model predicts the next required instrument for each working trocar of the surgeon separately, enabling enhanced context sensitivity of the surgery.

### Network architecture
The proposed model consists of ResNET50[51] and YOLOv8[52] for visual feature extraction followed by a Long-Term Context (LTContext) architecture[53] and a self-designed informed model. The full network architecture is visualized in Fig. 4.

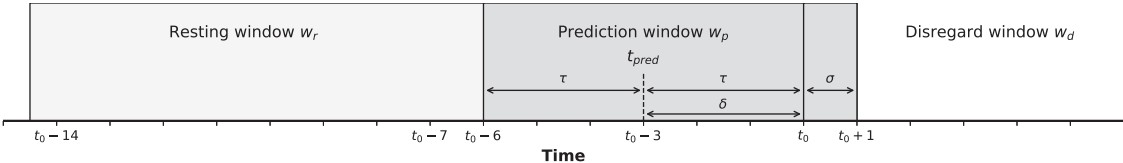

**Fig. 3 | Definition of resting, prediction, and disregard window.** For $\delta$ = 3 s, $\tau$ = 3 s and $\sigma$ = 1 s the prediction window ranges from $t_0 - 6$ s to $t_0 + 1$ s. In the prediction window, the next required instrument is predicted. Subsequently, the predicted instrument is picked up by the RSN and transferred to the surgeon, who inserts it into the corresponding working trocar. At time $t_0$, the instrument disappears from the laparoscopic video. A resting window precedes each prediction window to prevent possible false predictions and premature movements of the robot.

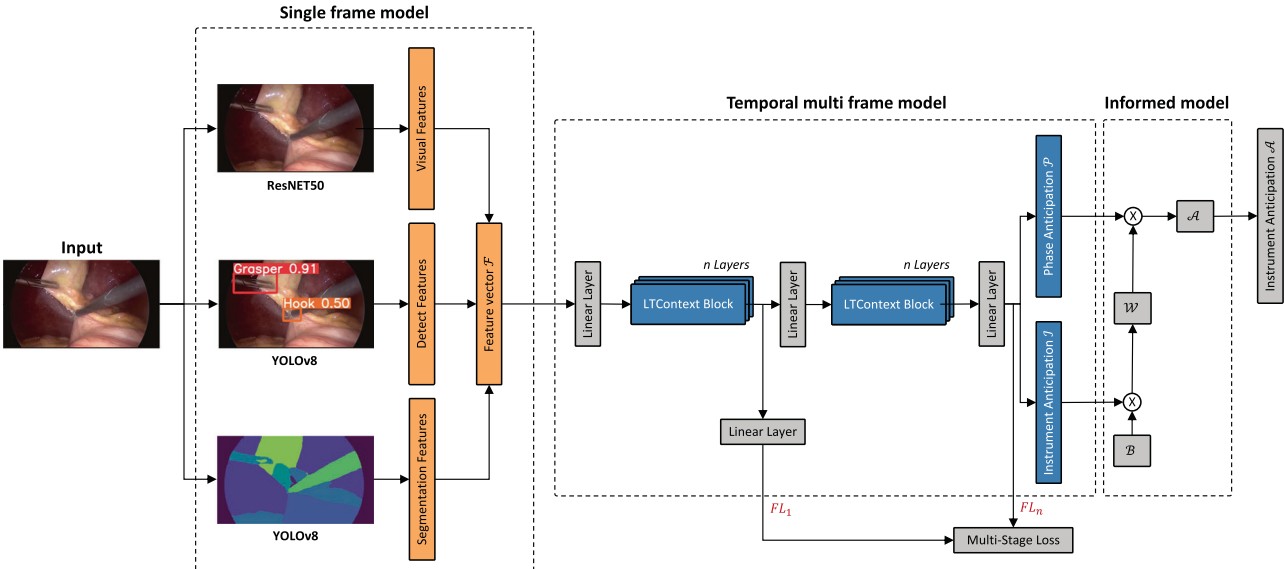

**Fig. 4 | Schematic network architecture of the three-stage model.** The architecture is composed of three feature extractor backbones, whose outputs are concatenated in the feature vector F. The feature vector is refined by a temporal multi frame model consisting of multiple stages of transformer-based LTContext blocks for anticipation of the next required instrument and the corresponding surgical phase. A subsequent informed model validates the prediction of the instrument using a phase compatibility matrix and adjusts uncertain predictions. The focal loss is calculated after each stage and aggregated for the joint training of the model.

Due to its success in prior studies[39], a ResNET50 was trained framewise without temporal context as a feature extractor from the videos on laparoscopic instrument, phase and action detection. The results of the feature extraction are the image features $V \in R^{2048}$ per image and the corresponding class probabilities $p(V) \in [0, 1]^{c_v}$ with $c_v$ as the number of classes. Using $c_v = 34$ classes (12 instruments, 11 phases and 11 actions) concatenated with the image features yields the visual feature vector $V \in R^{2082}$.

We trained a YOLOv8 network to recognize the laparoscopic instrument class and the spatial coordinates in terms of bounding boxes, resembling an instrument interaction module[34] to model the surgeons' intention during surgery. Instead of encoding the geometric relation between the grasper and other instruments, we concatenated the instrument class $c_i$, the bounding boxes ($x_i$, $y_i$, $w_i$, $h_i$), and the bounding box size $s_i = w_i \cdot h_i$ of each detected instrument, resulting in the detection feature vector $D \in R^{c_i \times 6}$.

To model the instrument's nearby anatomical structure, we trained another YOLOv8 network on organ and instrument tip and shaft segmentation. From the resulting segmentation masks, we calculated different image shape measurements such as eccentricity, extent, orientation, perimeter, and solidity for each detected organ, instrument tip or shaft class $c_s$. In addition, we concatenated the number of class areas, the proportion of a frame in the area, and the relative $x$ and $y$ position of an area with the image shape measurements, yielding the segmentation feature vector $S \in R^{c_s \times 9}$.

The feature vectors $V$, $D$, and $S$ are concatenated and sent into the temporal multi frame model for the temporal phase and instrument anticipation task. The LTContext model consists of several stages $S_{i...M}$ to refine the output of the first stage $S_1$. In contrast to ref. 54, we modified the model to be fully causal, allowing for intraoperative online deployment. After each stage $S_{1...M}$ we use the focal loss[55] (FL) to accurately train our model on the rarely occurring instrument changes. Moreover, we weight the focal loss with the scaling function shown in Eq. (1) to focus on predictions at the beginning of the prediction window, which results in well-timed triggering of the robotic motion. The variables $\beta$ and $\mu$ are hyperparameters. The degree of temporal scaling is adjustable by $\beta$, whereby $\mu$ defines the weighting of the disregard

window $w_d$.

$$scale(t) = \mathbb{1}_{w_p}(t)\left[\beta + \left(\frac{t - t_{w_{p,end}}}{t_{w_{p,end}} - t_{w_{p,start}}}\right)^2 \cdot (1 - \beta)\right] + \mathbb{1}_{w_r}(t)\left[\beta + \left(\frac{t_{w_{r,start}} - t}{t_{w_{r,end}} - t_{w_{r,start}}}\right)^2 \cdot (1 - \beta)\right] + \mathbb{1}_{w_d}(t) \cdot \mu$$

(1)

Multiplying the scaling function with the adjusted focal loss for multiclass classification and adding the losses of the individual stages $M$ yields the overall loss function in Eq. (2). For both the resting and the prediction window, we set the focusing parameter $\gamma = 2$, while for the disregard window we set $\gamma = 1$.

$$L(\hat{y}, y, t) = \frac{1}{M}\sum_{m}^{M} FL_m(\hat{y}, y, \gamma) \cdot scale(t)$$

(2)

To make the anticipation of an instrument even more reliable, we incorporate prior surgical knowledge by an informed model using an instrument phase compatibility matrix B. This binary matrix describes the natural usage of laparoscopic instruments in the respective phases of a surgery. By performing matrix multiplication with the instrument anticipation matrix I and instrument phase compatibility matrix B, we compute the weighted instrument matrix $W = I \times B$. The resulting weighted instrument matrix W assigns weights to each instrument based on its compatibility with other instruments, as determined by the binary relationships encoded in B. After normalizing the phase anticipation matrix P and the weighted instrument matrix W, we again perform matrix multiplication to obtain our final normalized instrument anticipation matrix $A' = P' \times W'$.

## Data sets

We created two non-publicly available data sets to train the RSN with real-world data. We recorded laparoscopic videos of 12 cholecystectomies at the University Hospital rechts der Isar with a resolution of 1920 × 1080 pixels at

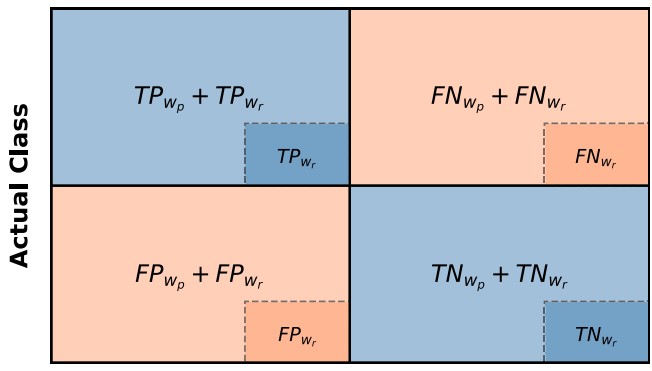

**Fig. 5 | Definition of the confusion matrices.** Confusion matrix of the resting window $C_{w_r} := \{TP_{w_r}, FP_{w_r}, FN_{w_r}, TN_{w_r}\}$ embedded in the confusion matrix of the prediction window $C_{w_p} := \{TP_{w_p}, FP_{w_p}, FN_{w_p}, TN_{w_p}\}$. The metrics are calculated based on both confusion matrices.

50 frames per second (fps). The 12 cholecystectomies were semantically annotated at 1 fps by medical experts with the annotation tool CVAT[56]. All videos provide pixel-wise annotation of laparoscopic instruments, while one video also includes pixel-wise annotation of the anatomical structure. This data set was used for learning the feature vectors $D$ and $S$.

The second data set, which we also recorded at the University Hospital rechts der Isar, includes 50 laparoscopic videos with a resolution of $1920 \times 1080$ pixels recorded at 25 fps or 50 fps. The data set contains the annotation of 11 phases, 12 instruments, and 11 actions. In addition, it provides information about the working trocar occupation of the laparoscopic instruments. The phases, instruments, actions, and trocar occupation were annotated by medical experts with the annotation tool ELAN[57]. The 50 videos were randomly split into training (68%), validation (16%), and test (16%) sets and used for learning both the feature vector $V$ and the temporal multi frame model. The videos were sub-sampled to 1fps and resized to a spatial resolution of $224 \times 224$ pixels to reduce computational costs.

The data were collected with ethical approval and written informed consent from the participants. Approval was granted by the Ethics Committee of the University Hospital rechts der Isar (No. 337/21 S). This study was performed in line with the principles of the Declaration of Helsinki. The authors also affirm that they have received written informed consent from all individuals for the publication of the images contained in this manuscript.

### Model training and deployment
The YOLOv8 feature extractors were trained on the first data set while the ResNET50 was trained on the second data set. The temporal model was also trained on the second data set using a hyperparameter sweep. The best results for the task of surgical instrument anticipation were achieved for the number of stages $S = 1$ and layers $n = 8$ using the Adam optimizer with an initial learning rate of $5e - 5$ for 550 epochs. The warm-up amounted to 70 epochs. We provide the test results of the model that performed best on the validation set. The batch size is identical to the length of each video. The model was implemented in PyTorch and trained on three NVIDIA RTX A6000s. For intraoperative online deployment, we built an inference pipeline using NVIDIA Clara Holoscan. The model outputs are transferred from a NVIDIA AGX development kit to the robot via the robot operating system (ROS).

### Evaluation metrics
To comprehensively measure the results of the predictions, we consider each resting window followed by a prediction window as one sequence. We set the label of the sequence equal to the ground truth of the prediction window. For each sequence, we check whether the first prediction $\hat{y}(i, \alpha) \neq y_0$ matches the corresponding label $y(i, \alpha)$ and track the position of the occurrence

of the prediction. For comparing $\hat{y}(i, \alpha)$ against $y(i, \alpha)$, the confusion matrix for the prediction window $C_{w_p}$ and the resting window $C_{w_r}$ are computed. By embedding $C_{w_r}$ in $C_{w_p + w_r}$, we can evaluate the performance of the ML system in a differentiated way and at the same time evaluate how well the integration of the robotic system can be realized (see Fig. 5). Based on the confusion matrix we deploy four different metrics suitable for classification tasks, namely precision, recall, accuracy and F1 score[58]:

$$Precision = \frac{(TP_{w_p} + TP_{w_r})}{(TP_{w_p} + TP_{w_r}) + (FP_{w_p} + FP_{w_r})} \quad (3)$$

$$Recall = \frac{(TP_{w_p} + TP_{w_r})}{(TP_{w_p} + TP_{w_r}) + (FN_{w_p} + FN_{w_r})} \quad (4)$$

$$Accuracy = \frac{(TP_{w_p} + TP_{w_r}) + (TN_{w_p} + TN_{w_r})}{(TP_{w_p} + TP_{w_r}) + (TN_{w_p} + TN_{w_r}) + (FP_{w_p} + FP_{w_r}) + (FN_{w_p} + FN_{w_r})} \quad (5)$$

$$F1 = \frac{2 \cdot (TP_{w_p} + TP_{w_r})}{2 \cdot (TP_{w_p} + TP_{w_r}) + (FP_{w_p} + FP_{w_r}) + (FN_{w_p} + FN_{w_r})} \quad (6)$$

The evaluation metrics are computed class-wise, for each instrument and working trocar individually, and then averaged over all videos (operations) in the test set, yielding the weighted-averaged precision (wAP), recall (wAR), accuracy (wAA) and F1 score (wAF1). The weighted-averaged metrics are calculated by taking the mean of all per-class scores while considering each class's support, whereby the support refers to the number of actual occurrences of the class in the test data set. In Supplementary Table 3 and Supplementary Figs. 11 and 13, we also provide the macro average metrics[59] per class as well as a multiclass confusion matrix for each working trocar separately. In this context, we neglect predictions in the disregard window $w_d$ as these predictions do not harm the surgical workflow but would pose unnecessary robotic movements.

### Statistics and reproducibility
The model performance was assessed using the metrics described such as weighted-averaged precision, recall, accuracy and F1 score. Hyperparameter tuning was conducted using Bayesian search. The data set for learning the instrument anticipation task consisted of 50 samples, which were randomly split into training and testing sets. The training set contained 34 samples, while the testing set contained 8 samples. A validation set of 8 samples was used for hyperparameter tuning. The temporal multi frame model was trained 10 times using different random seeds to initialize the model parameters assessing stability and reliability of the model performance.

### Reporting summary
Further information on research design is available in the Nature Portfolio Reporting Summary linked to this article.

### Results
The evaluation of the instrument prediction system was performed on eight surgeries of laparoscopic cholecystectomies, which took place between 2019 and 2022 at the University Hospital rechts der Isar of the Technical University of Munich. The surgeries lasted an average of 30 min 19.50 s ± 10 min 58.45 s and contained 24.13 ± 2.89 instrument changes. Considering the two working trocars providing intra-abdominal access, the instrument changes are distributed as follows: The instrument in the left trocar was changed 4.13 ± 0.78 times on average, while the instrument in the right trocar was changed 20.0 ± 3.0 times.

**Table 1 | Evaluation metrics for the two different temporal model types: MS-TCN and LTContext**

| Model | wAP | wAR | wAA | wAF1 |
|---|---|---|---|---|
| MS-TCN | 74.51 ± 1.86 | 69.59 ± 1.45 | 87.81 ± 0.79 | 70.29 ± 1.65 |
| LTContext | 77.15 ± 1.93 | 70.31 ± 0.96 | 88.32 ± 0.64 | 71.54 ± 1.07 |

The evaluation metrics are computed class-wise, for each instrument and working trocar individually, and then averaged over all surgeries in the test set, yielding the weighted-averaged precision (wAP), recall (wAR), accuracy (wAA), and F1 score (wAF1). The weighted-averaged metrics are calculated by taking the mean of all per-class scores while considering the number of actual occurrences of each class in the test data set. The averaged metrics over tenfold are reported (%) with the corresponding standard deviation ( ± ).

**Table 2 | Drop of the weighted-averaged F1 score as a result of the perturbation of the individual feature extractor backbones**

| Ablative module | − ΔwAF1 |
|---|---|
| Visual feature module | 33.39 ± 4.46 |
| Detection feature module | 6.77 ± 2.54 |
| Segmentation feature module | 0.43 ± 0.56 |

The highest drop is achieved by the visual feature module, indicating that the module extracts the most important information for the prediction. The detection feature module provides a smaller impact on the model performance, while the segmentation feature module contributes only marginally to a better prediction of the surgical instruments in terms of the F1 score. The drop of the averaged F1 score over 10 folds is reported (%) with the corresponding standard deviation ( ± ).

## Comparative methods

In Table 1, we compare the LTContext architecture[53] with a Multistage Temporal Convolutional Network (MS-TCN)[54] which is typically used for surgical instrument anticipation tasks[34,35], both followed by an informed model. The models differ in the incorporation of temporal information in their predictions. Through the informed model, we include prior surgical knowledge into our computational pipeline, allowing for improved instrument prediction for each working trocar separately.

We report the wAP, wAR, wAA, and wAF1 for both of our approaches considering the support of each class. The support refers to the number of actual occurrences of the class in the test surgeries. As illustrated in Table 1, the MS-TCN approach is outperformed by 3% in terms of weighted-averaged precision. The weighted-averaged recall and accuracy of the LTContext approach are also slightly higher at 70.31% and 88.32% than the scores of the MS-TCN approach. The LTContext model also achieves a slightly higher F1 score (71.54%) than the MS-TCN model (70.29%). Thus, the derived LTContext architecture is able to forecast 71.54% of instrument changes, ensuring seamless integration into the surgical workflow by delivering these predictions in advance for instrument preparation. These results are obtained exclusively on the basis of laparoscopic video input.

Supplementary Figs. 11 and 12 show that some instruments are more predictable than others. In particular, the instrument changes of the grasper (F1 95.66 ± 2.19%) and the irrigator (F1 95.73 ± 2.13%) in the left working trocar can be predicted accurately. For the right working trocar the model manages to predict the majority of required instrument handovers correctly, such as the biopsy forceps (F1 68.47 ± 2.34%) and the coagulation suction tube (F1 77.77 ± 3.32%), which are often used to prepare tissue structures, as well as the clipper (F1 75.71 ± 2.70%) and drain (F1 77.06 ± 2.16%).

However, the prediction of surgical instruments such as the scissors (F1 34.48 ± 10.23%) is more challenging, as the number of clips depends on the surgeon's preferences and the patient's anatomical structure. Usually, the cystic duct and the cystic artery are clipped twice on the patient side and once on the specimen side after having achieved the critical view of safety. In the cases of anatomical norm variants or unclear anatomical structure, it may be necessary to apply additional clips. This complicates the prediction of the scissors, which is responsible for the subsequent dissection of the two structures.

The drain and the retrieval bag, both single-use devices, achieve better anticipation values, although they are only needed once per intervention. The drain as a phase-related instrument, usually inserted at the end of an operation, reaches a higher F1 score as stated above. The retrieval bag in contrast is more difficult to anticipate (F1 54.11 ± 6.54%), as the time point of packing of the resectate is surgeon-dependent, which is reflected by a higher false positive value. In other words, the prediction system is somewhat more likely to predict nothing than the retrieval bag.

The most challenging instrument to anticipate is the irrigator (F1 16.63 ± 9.20%), which is inserted sporadically in the right working trocar in case of bleeding or excessive smoke development due to coagulation. Throughout the entire test surgeries, the irrigator is only requested three times for the right trocar, which makes a reliable prediction unachievable.

Apart from the irrigator and the scissors, whereby the former can be regarded as edge case in laparoscopic cholecytectomies, the results show that the instrument prediction system reliably provides anticipations for the vast majority of instrument handovers, enabling timely instrument delivery by the RSN.

## Effect of different feature extractors

We conducted ablation studies to assess the impact of the three different feature extractor modules in our single-frame model. For this, we used the perturbation-based method occlusion[60] on module level by replacing the input features of each feature extractor with zeros. We reported the magnitude changes in the model output regarding the F1 score −ΔwAF1 in Table 2.

The feature extraction module provides the largest drop in terms of the F1 score (−Δ33.39 ± 4.46%). The detection feature module achieves a smaller decrease (−Δ6.77 ± 2.54%), while the F1 score changes only slightly when perturbing the segmentation feature module (−Δ0.43 ± 0.56%). In other words, the features of the segmentation backbone model provide the least added value for the prediction of the next required instrument. As the perturbation method was conducted with the value zero, which is not informationless, it may introduce a bias in the system's interpretation of the data, potentially affecting the ablation scores. Nevertheless, the results clearly show that the visual feature extraction module provides the greatest contribution, although its underlying data set requires the least annotation effort.

## Discussion

Surgeries take place in challenging environments, and surgeons today must deal with a variety of human factors, NTS-related challenges and infrastructural challenges[16,17], the latter of which particularly affect the quality and efficiency of surgical workflows (see Fig. 1). Staff shortages, lack of prior knowledge and lack of process standardization were rated as particularly frequent and prevalent challenges in the survey among surgeons we conducted from August until October 2023 in German clinics. The passing of surgical instruments and sterile goods from the scrub nurse to the surgeon is an important and recurring process step in all operating phases which often requires a verbal request by the surgeon[9]. The aforementioned challenges impede the nonverbal communication between the scrub nurse and the surgeon, despite research having already demonstrated the significance of nonverbal communication in such settings[10,11]. While process optimization and nonverbal communication training are already widely studied[11,61], this study demonstrates how RSNs can be trained to anticipate and hand over required surgical instruments to overcome infrastructural challenges such as staff burden, improve nonverbal communication and enable seamless handover procedures that avoid downtime.

In our study, we developed an RSN that aims to address infrastructural challenges and improve the efficiency of surgical workflows by predicting and handing over required surgical instruments based on laparoscopic video signals. The core contribution lies in the network architecture that enables the model to anticipate the next required surgical instrument. Our RSN uses a unique combination of a three-stage algorithm including a single-frame-, temporal multi frame-, and an informed model. This setup is specifically

tailored to the task of surgical instrument anticipation, which requires accuracy as well as fast-paced adaption to new situations during surgery.

The instrument prediction system anticipates surgical instruments with an overall F1 score of 71.54%. Besides prediction accuracy, this score considers that the RSN has enough time to hand over the predicted instrument to the surgeon as needed. Instruments of the left working trocar are more reliably predicted (F1 >95%) as these instrument changes are primarily phase-related and take place at phase transitions. Since instruments in the right working trocar are exchanged more dynamically, the difficulty of anticipation of these instruments is higher. For the scissors and clipper, challenges such as the large influence of the surgeon's preference on temporal utilization lead to more false negative errors as outlined in Supplementary Figs. 11 and 12, as the prediction system struggles to correctly estimate the desired order of clipper and scissors. However, the prediction system can anticipate the majority of handovers, while standardized routine tasks can be anticipated more reliably. The results show that the ML-based instrument prediction system enables RSNs to complement human capabilities to free up capacity for situations that require a high mental workload. Based on these findings, we will now discuss how the developed instrument prediction system contributes to improving surgical workflows. In the following, we refer to RSNs that are equipped with the developed instrument prediction system.

First, one of the most notable impacts of the RSN is its capacity to reduce verbal communication in the operating room. By accurately predicting and handing over the required surgical instruments, the RSN acts as a mind reader, addressing 71.54% of instrument requests in the data set containing recordings of real operations without the need for verbal cues. This reduction in verbal communication does not only improve efficiency; it also contributes to a quieter, more focused environment, which is essential for high-stakes and high-quality surgical procedures.

Second, by ensuring that required surgical instruments are anticipated and prepared in advance, the RSN enables surgeons to maintain a steady and standardized flow in their surgical procedures which positively impacts the overall efficiency. In addition, the informed model provides the RSN with knowledge about standardized surgical procedures which reduces the likelihood of variability and error, leading to more predictable and streamlined workflows.

Third, the RSN presents a promising solution to the growing concern of staff shortages. By autonomously handling most instrument anticipation and handover tasks, the RSN can effectively supplement the work of human scrub nurses. Although benefits such as long-term cost savings may occur once trained RSNs are deployed in multiple clinics and the deployment scales, one of the main contributions of the presented approach is that it can compensate for what is increasingly lacking in the operating room—experienced scrub nurses—and thus ensure that surgeries can continue to be performed efficiently and safely.

One of the primary limitations of the presented RSN is its exclusive reliance on laparoscopic video input. This focus on intra-abdominal events means that extra-abdominal events, which can significantly influence surgical procedures, are not accounted for in the RSN's instrument anticipation task. While incorporating extra-abdominal recordings would enhance the system's comprehensiveness, this approach faces substantial hurdles, particularly regarding data protection and privacy concerns.

While instrument handover is a common and relevant process in surgeries worldwide, process standards differ between regions and individual clinics. RSNs need to be trained for different standards and continuously adapt to ongoing developments. Maintenance and retraining of models, as well as the further development of data sets, represent important challenges for future research.

Lastly, a notable limitation in our current development of the RSN is its training on a data set derived from a singular type of surgical procedure, performed across 34 surgeries. This limited scope restricts the RSN's ability to generalize across a wide range of surgical contexts. To establish RSNs as versatile tools capable of assisting in various surgeries, a more diverse and extensive training data set is essential. Moreover, it is important to

acknowledge that surgical procedures do not always adhere to standard protocols. Unforeseen circumstances, such as bleeding incidents, introduce complexities that our current RSN may not be adequately equipped to handle (irrigator F1 16.63 ± 9.20%). Training RSNs to effectively respond in these edge cases will be a critical task for future research to enable RSNs to further contribute to patient safety. Moreover, most laparoscopic interventions share a high level of standardization and our approach could be transferable to other laparoscopic procedures following additional data collection and annotation. However, the transferability to less standardized procedures is limited and differences in surgical procedures between different clinics present challenges to the scalability of the presented approach. For example, while the use of a drain during laparoscopic cholecystectomies is an internal standard practice at the clinic where surgeries were recorded, it is not a universally applied approach.

Moving forward, it is essential to delve into the broader implications of integrating RSNs into OR teams, particularly focusing on the dynamics of collaboration within these hybrid human-robot teams. It is imperative to explore how surgeons and scrub nurses perceive and adapt to working alongside robotic counterparts, as these perceptions can significantly influence the effectiveness of the RSN's integration and the team's performance as a whole.

As we progress towards the integration of RSNs in OR teams, it is necessary to establish interim solutions and strategies for a safe path to translation. First, we suggest an initial deployment of RSNs in less complex and lower-risk phases of surgical procedures. This cautious approach allows for the system to be tested and refined in a controlled environment, ensuring safety and reliability before moving to broader applications. To continuously improve the accuracy and performance of RSNs' instrument prediction system, ongoing training during operation is vital. By integrating feedback and new data collected during surgeries, the system can evolve and adapt, reducing errors and increasing its predictive capabilities. In addition, we envision an interim deployment strategy where a human scrub nurse oversees a RSN in the roll-out phase, stepping in to provide guidance or intervene in critical situations. This hybrid approach does not only leverage the strengths of both humans and machines, but also adds an extra safety layer during the transition phase. Given that ML systems are prone to errors[62], especially in the anticipation of infrequently used instruments, the implementation of mutual supervision mechanisms becomes critical. Furthermore, we propose the merging of voice input and predictive capabilities in future studies as a means to facilitate real-time corrections and adjustments by surgeons or scrub nurses overseeing the system. In addition, by allowing the RSN to visually or audibly communicate the predicted instruments to the surgeon prior to the actual handover, the system provides the opportunity to prevent delays and ensure the continuity of the surgical workflow. This ensures that any discrepancies or malfunctions in the RSN's operations can be promptly identified and rectified, thereby safeguarding the efficacy of surgical procedures and maintaining the highest standards of patient care.

In addition, the RSN was trained on clinic-specific data and solely on annotated video recordings of laparoscopic cholecystectomies, which is a highly standardized procedure. In order to promote the better dissemination of standardized OR processes and to train RSNs for use in various clinics, the development of a cross-clinic data platform that enables access to more diverse data for the development of ML systems based on federated learning approaches would be of great relevance in the future[63]. The study also underscores the importance of future standardization of surgical procedures for the development and implementation of reliable ML systems in the OR.

In general, results from our study demonstrate that the RSN can autonomously manage 71.54% of instrument anticipation and handover tasks, which not only highlights their potential to streamline surgical workflows but also opens avenues for future applications in educational settings. An intriguing prospect for future research is the utilization of RSNs as training tools for novice human scrub nurses. Future research should thus explore the design and implementation of RSNs for educational purposes

and evaluate their effectiveness in enhancing the skill set of novice scrub nurses and residents.

Overall, the presented instrument prediction system is capable of enabling the RSN to become a mind reader in order to anticipate and hand over the vast majority of required surgical instruments, reducing the need for verbal communication and streamlining the instrument handover process.

## Data availability
The main data supporting the results of this study are available within the paper and its Supplementary Information. Source data for Fig. 1 can be found in Supplementary Data 1 and source data for Tables 1 and 2 can be found in Supplementary Data 2. The data sets generated and analyzed during the current study are not publicly available due to restrictions related to privacy concerns for the research participants but are available from the corresponding author on reasonable request.

## Code availability
The underlying code for this study is available at https://github.com/ResearchgroupMITI/instrument-anticipation and is provided in Zenodo[64]. The following packages were used: Python 3.10.13, PyTorch 2.0.0, Torchvision 0.15.2, Numpy 1.26.0, Pandas 2.1.1, Matplotlib 3.8.0, Scikit-image 0.20.0, Scikit-learn 1.3.0, Tensorboard 2.12.1, and Wandb 0.15.12.

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

## Acknowledgements
The authors acknowledge funding support by the Bavarian Ministry of Economic Affairs, Regional Development and Energy (StMWi) (grant number: DIK0372). The authors would also like to thank NVIDIA for the Clara AGX development kit donation.

## Author contributions

L.W., S.J., and L.M. developed and implemented the proposed methodology. L.W., S.J., L.M., and C.M. designed and evaluated all technical experiments. L.W., L.M., and D.W. consulted S.J., A.K., and F.H. on the design of the proposed model. L.W., M.B., H.F., and D.W. collected and curated the data sets. S.J., D.W., A.J., P.B., and M.B. acquired survey participants and evaluated the survey. D.W. and A.K. defined the clinical goal of the study. L.W., S.J., C.M., L.B., S.K., and D.W. generated images and figures of the manuscript. L.W., S.J., L.M., P.B., and S.K. wrote different parts of the manuscript. All authors reviewed the manuscript.

## Funding

## Competing interests
The authors declare no competing interests.

## Additional information

[1]Technical University of Munich, TUM School of Medicine and Health, Klinikum rechts der Isar, Research Group MITI, Munich, Germany. [2]Technical University of Darmstadt, Software & Digital Business Group, Darmstadt, Germany. [3]Technical University of Munich, TUM School of Medicine and Health, Klinikum rechts der Isar, Department of Surgery, Munich, Germany. [4]Technical University of Munich, TUM School of Computation, Information and Technology, Chair of Robotics, Artificial Intelligence and Real-Time Systems, Garching, Germany. [5]Technical University of Munich, Munich Institute of Robotics and Machine Intelligence, Munich, Germany. [6]These authors contributed equally: Lars Wagner, Sara Jourdan. [7]These authors jointly supervised this work: Peter Buxmann, Alois Knoll, Dirk Wilhelm. ✉e-mail: lars.wagner@tum.de

