## [Peer Review File · Communications Medicine]

Reviewers' comments:

Reviewer #1 (Remarks to the Author):

Many thanks for the opportunity to review this article which presents a novel and interesting concept. While robotic assistance has been debated within the literature, the concept of a robotic scrub nurse is thought provoking and I enjoyed reading the paper.

The paper is well written and well structured and easy to follow. I commend the authors in conducting a survey as part of the introduction giving solid justification for the conduct of the study.

I have a few queries:

- a) Why did the authors choose lap chole? Due to availability of video or due to standardisation or previous work? It would be good to present what difficulties transferring this to a less standardised procedure would be envisaged?
- b) From a clinical point of view, what is the time/efficiency saving estimated - is there enough clinical benefit for surgeons to adopt this technology? I think there needs to be more emphasis on the translational potential of this - ie could I not just have the instruments in front of me for the price of this significant technology.
- c) Do the authors mean a drain when they state drainage? Does this mean the system predicts the need for a drain - would you be able to clarify how this works. i.e. I would not routinely place a drain for a laparoscopic cholecystectomy unless there is significant soiling/collection/risk of bile leak
- d) What are the parallels between the instruments that the system finds trickiest to predict vs those of human scrub nurses? I suspect fairly similar. If so, what is the merit for the robotic system
- e) Would be good to present the perceived path to translation - is this a feasible product in the OR and how do we get there?

Reviewer #2 (Remarks to the Author):

This paper proposes a method for reading the surgeon's intention, which is an important element of the user interface for RSNs that hand surgical instruments to the surgeon.

Given the limitations of the information used, surgical technique, and surgeon dependence, this accuracy is not sufficient, but it is a challenging method and this paper is a useful reference for the future.

Overall the paper is well written and organized, and expected to have a large readership. It is recommended that the following points be strengthened for better revision.

Comments:

In the introduction, the results in Figure 1, while an interesting topic, do not adequately demonstrate significant differences, nor do they clarify the author's claim of RSN requirements. It would be better written in a more convincing manner using other data in the Appendix.

There are no differences between the MS-TCN and LTContext models that were tried as research methods, and we do not believe that their use in this method alone is sufficient to achieve the

objective. Although you indicated the limitations of the study in the discussion, more specific information and solutions should be discussed. And more, it would also be very interesting to discuss what percentage of reading accuracy the surgeon would be satisfied with while the training data is highly dependent on the technique and surgeon.

In addition, it would be interesting to see what the learning results from other data sets would be in future studies, and it would be good to mention the standardization of the surgical process, etc., to suggest that this could revolutionize existing treatment procedures.

The first of the most significant effects of RSNs described in the discussion, the reduction of verbal communication, may also change the efficiency of some parties.

Time variations due to misidentification and emergencies would have a significant impact on efficiency. The mention of the safety for this system is insufficient, and it would be good to also mention methods for dealing with misidentification, etc.

As for Equations 3 through 6, they are common sense statistical formulas and should be removed unless there is something special about them. Or in supplemental material section.

Reviewer #1:Comment 1

Why did the authors choose lap chole? Due to availability of video or due to standardisation or previous work? It would be good to present what difficulties transferring this to a less standardised procedure would be envisaged?

Response:

Thank you for your insightful question regarding our choice of laparoscopic cholecystectomy for the application of our machine learning (ML) model in our study. First, having access to a significant number of laparoscopic cholecystectomies was essential to capture our video data. This allowed us to train our ML model on a large and comprehensive data set. Second, the involvement and availability of medical experts in laparoscopic cholecystectomies were crucial for high-quality data annotation and interpreting the model's output. Their domain knowledge ensured accurate instrument identification and procedural annotations. Lastly, a laparoscopic cholecystectomy is a highly standardized procedure, which significantly benefits the development and application of ML-based decision support systems. A standardization enables the derivation of reliable patterns from video recordings, contributing to the model's predictive accuracy. Most laparoscopic interventions share a high level of standardization, suggesting that our approach may be transferable to other procedures following additional data collection and annotation as well as the potential scalability of our ML-based instrument prediction system.

Regarding the transferability to less standardized procedures, we acknowledge this as a noteworthy challenge. The high degree of standardization in laparoscopic cholecystectomies make it an ideal case study for implementing and evaluating ML technologies in surgery. However, we recognize the need for surgical procedure standardization to facilitate the broader integration of ML technologies into the operating room.

Change(s):

In our discussion section, we now explicitly address the challenges associated with applying our developed approach to less standardized procedures. This reflection not only highlights the current limitations but also emphasizes the necessity for ongoing standardization efforts in surgery to maximize the benefits of ML technologies. We appreciate your thoughtful questions and hope our response clarifies the rationale behind our selection of the surgical procedure and the potential transferability of our results.

Comment 2

From a clinical point of view, what is the time/efficiency saving estimated - is there enough clinical benefit for surgeons to adopt this technology? I think there needs to be more emphasis on the translational potential of this - ie could I not just have the instruments in front of me for the price of this significant technology.

Response:

Thank you very much. We appreciate the opportunity to discuss the clinical benefits and potential efficiency/ cost improvements provided by the developed instrument prediction system for RSNs. It's important to clarify that the primary benefit of the RSN is not necessarily quantified through direct time savings, but rather through compensating for what is increasingly lacking in the operating room - experienced scrub nurses - and thus ensuring that surgeries can continue to be performed efficiently and safely in the face of these increasing staff shortages. While the initial cost of implementing such advanced technology might appear significant, it is crucial to consider the long-term implications of addressing infrastructural challenges, particularly the shortage of skilled personnel.

While cost savings are not the primary motivation for developing the instrument prediction system to support RSNs, the scalability of the RSN approach offers significant potential for cost effectiveness. Once surgical processes are adequately standardized, the trained instrument prediction system can

empower multiple RSNs across clinics without significant additional development costs. This scalability could lead to long-term cost savings and improvements in surgical outcomes across a broad range of procedures.

In addition, the advantage of the RSN over having instruments pre-positioned is in enhancing process quality and maintaining an uninterrupted surgical workflow. The RSN enables surgeons to keep their focus on the surgical field or the monitor in the OR without the need to divert attention to instrument selection, mimicking the “perfect assistant” or “mind reader” that can anticipate the surgeon’s needs without verbal communication.

Lastly, we believe in a collaborative approach of machines and humans in a sense, that we aim not for replacing scrub nurses completely, but to assign them for more complex and demanding surgeries, while less complex and standardized surgeries could be given in the hands of our robotic system.

In summary, the value proposition of the RSN extends beyond immediate cost savings to address critical infrastructural challenges such as staff shortages, enhance process quality, and maintain uninterrupted surgical workflows. This, we believe, represents a compelling case for the adoption of ML technologies in the OR.

Change(s):

We would like to continue to focus on the following contributions that the RSN provides instead of focusing on potential cost savings: First, the capacity to reduce verbal communication by acting as a mind reader, second, the RSNs potential to standardize procedures and improve efficiency by anticipating instruments in advance, lastly, the opportunity to alleviate staff shortages. We urgently need solutions to address the lack of experienced scrub nurses. We now discuss this aspect in more detail in our discussion section. However, we now also mention in our discussion the potential to scale trained models and deploy them across various clinics which could lead to long-term cost savings.

Comment 3

Do the authors mean a drain when they state drainage? Does this mean the system predicts the need for a drain - would you be able to clarify how this works. i.e. I would not routinely place a drain for a laparoscopic cholecystectomy unless there is significant soiling/collection/risk of bile leak

Response:

Thank you for highlighting the terminology and practice surrounding the use of a drainage in our study. When referring to drainage, we indeed mean the use of a drain, which, as you correctly pointed out, is not routinely placed in laparoscopic cholecystectomies. Our mention of drain in the manuscript reflects an internal standard practice within our clinic, where drains have been routinely used in the context of laparoscopic cholecystectomies and thus in all our recorded surgeries that we used to train the ML model. We acknowledge that this approach is not universally applied. The routine use of drain in our procedures, while consistent with our internal standards, does indeed present a limitation in the generalizability of our findings, especially concerning the predictive model’s application in settings where the placement of a drain is not standard practice. However, consideration of the drain is necessary for a reliable model training in our setup.

Change(s):

Considering your valuable feedback, we will clarify this aspect in our study to indicate that the prediction of a need for a drain by our system is based on internal clinical practices. We consider this aspect in the limitations when discussing challenges for the scalability of the trained model across various clinics.

Comment 4

What are the parallels between the instruments that the system finds trickiest to predict vs those of human scrub nurses? I suspect fairly similar. If so, what is the merit for the robotic system

Response:

Thank you for your interesting query regarding the comparison between the instruments our system finds challenging to predict and those that human scrub nurses might struggle with. As indicated by one of our leading surgeons, difficulties arise predominantly when multiple instrument choices are possible. This challenge applies for both the ML system and human scrub nurses. However, the primary goal of the instrument prediction system and the RSN is not to replace personnel by outperforming human capabilities. Instead, there is a shortage of experienced scrub nurses, and surgeons' needs are less and less anticipated. The use of RSNs can counteract staff shortages but will initially have to be deployed under human oversight.

We must also note that we do not have a data set of erroneous human handover incidents, which limits our ability to make a quantitative comparison between our system's performance and that of human scrub nurses. This limitation notwithstanding, the merit of the robotic system lies in its ability to facilitate a seamless surgical workflow through non-verbal communication, compensating for the lack of (experienced) scrub nurses capable of such anticipation.

Comment 5

Would be good to present the perceived path to translation - is this a feasible product in the OR and how do we get there?

Response:

We appreciate your interest in the translational pathway of the presented RSN into a practical, operational tool within the OR. In our discussion on future research directions, we highlight several critical aspects essential for this transition, including change management for hybrid surgical teams and the necessity for human oversight due to the inherent error-susceptibility of ML systems. Building on this foundation, we now delve deeper into additional aspects and strategies crucial for the safe and reliable system's integration in a clinical setting.

Change(s):

Firstly, we now suggest an initial deployment of the system in less complex and lower-risk phases of surgical procedures. This cautious approach allows for the system to be tested and refined in a controlled environment, ensuring safety and reliability before broader application.

To continuously improve the accuracy and performance of the model, ongoing training during operation is vital. By integrating feedback and new data collected during surgeries, the system can evolve and adapt, reducing errors and increasing its predictive capabilities.

In addition to technological advancements, the role of experienced human scrub nurses remains paramount. We envision a model where a human scrub nurse oversees the RSN in the roll-out phase, stepping in to provide guidance or intervene in critical situations. This hybrid approach not only leverages the strengths of both human and machine, but also adds an extra safety layer during the transition phase. Furthermore, we propose to merge the predictive and voice input capabilities to facilitate real-time corrections and adjustments by surgeons or scrub nurses who still oversee the system, as our system also features voice control, which was not explicitly mentioned in this article.

By addressing these aspects, we aim to outline a clear and feasible path towards integrating our RSN system into the OR. We aim to address the technical readiness of the system but also the operational and safety considerations and the concerns regarding surgical team dynamics which are essential for successful adoption and implementation in clinical practice.

Reviewer #2:Comment 1

In the introduction, the results in Figure 1, while an interesting topic, do not adequately demonstrate significant differences, nor do they clarify the author's claim of RSN requirements. It would be better written in a more convincing manner using other data in the Appendix.

Response:

Thank you for your feedback on the presentation of Figure 1 and its implications for RSN requirements. We acknowledge the need for clarity in how we present and interpret the survey results within our introduction. The selected survey results presented in Figure 1 are intended to highlight the infrastructural challenges in German clinics which were identified by both novice and experienced surgeons. The goal was not to measure differences between groups but to underscore the broad consensus on the importance of addressing these challenges, which motivates the exploration of RSNs as a potential solution.

In response to your comment, we have revised the introduction to clarify that the survey results highlight the need for innovative approaches to these infrastructural challenges, rather than directly defining RSN system requirements. This adjustment ensures our presentation aligns with the survey's intent and addresses your concerns about potential misleading implications. We have also considered presenting other results from the survey, which are currently listed in the Appendix. However, we agree with Reviewer 1 that the results on infrastructural challenges best fit the potential contributions of an RSN and e.g., human factors related challenges remain more interesting for future research dealing with hybrid human-robot surgical teams. We appreciate your constructive feedback, which has been highly valuable in refining our manuscript to convey our research motivation and objectives more accurately.

Change(s):

We revised the introduction of our study where we reference selected results from the conducted survey among surgeons from different German clinics. We now clearly describe that the survey only outlines the severity of infrastructural challenges. Rather than directly inferring the need to develop an RSN or design requirements, we now interpret these results as motivation to derive innovative solutions, one of which could be the integration of ML and robotic technologies in the OR. We further hope to inspire future research with this current surgeons' assessment of the clinical challenges.

Comment 2

There are no differences between the MS-TCN and LTContext models that were tried as research methods, and we do not believe that their use in this method alone is sufficient to achieve the objective. Although you indicated the limitations of the study in the discussion, more specific information and solutions should be discussed. And more, it would also be very interesting to discuss what percentage of reading accuracy the surgeon would be satisfied with while the training data is highly dependent on the technique and surgeon.

Response:

Thank you for your feedback on the limitations of the developed models and their potential to enable a RSN to reliably predict and handover surgical instruments. We see similarities in your comment and comment 5 of Reviewer 1 who asks for a translational path to integrate RSNs into the OR.

In response, we have refined our discussion to better address limitations and outline strategies for the safe deployment of RSNs in the future. First of all, we now emphasize that continuous improvement of models is necessary and that human feedback should be incorporated. Moreover, additional data from real-world clinical settings can be collected during use and could support the systematic improvement of the model's accuracy and adaptability over time.

In addition, we now propose a cautious initial deployment of RSNs during less complex and lower-risk surgical phases. This strategy allows for controlled testing and refinement, ensuring the system's readiness before wider application. We envision a hybrid operational deployment strategy where an experienced human scrub nurse oversees the RSN in the roll-out phase. In addition to technological enhancements, we highlight the potential integration of voice input capabilities to allow for immediate corrections and adjustments by the overseeing medical professionals. By outlining these strategies and considerations, we aim to demonstrate a comprehensive pathway toward the improvement and successful integration of RSNs into the OR room.

Regarding the specific query about reading accuracy, one of our collaborating surgeons suggested that an RSN system that achieves less than human accuracy could still provide significant value, especially in contexts where the alternative might be the cancellation of surgeries due to staffing shortages. The surgeon estimated a desirable accuracy threshold of approximately 90% for long-term deployment. However, we have chosen not to explicitly cite this figure in our study, considering it reflects the perspective of a single surgeon and is based on a hypothetical scenario rather than empirical evidence. We believe that setting such a specific benchmark at this stage could be premature and might not comprehensively reflect the varied contexts and complexities of surgical procedures.

Change(s):

The main changes include adjustments in the discussion of the study. Here we outline that the RSN should not simply be used as a replacement for experienced scrub nurses. Firstly, the performance should be continuously improved during operation as described above and, secondly, interim solutions are required in which, for example, several RSNs are supervised by human scrub nurses, especially in high-risk phases of surgical procedures. In the future, the surgeon should also be able to use voice input to provide individual preferences as feedback to the system. All changes are highlighted in the attached document.

Comment 3

In addition, it would be interesting to see what the learning results from other data sets would be in future studies, and it would be good to mention the standardization of the surgical process, etc., to suggest that this could revolutionize existing treatment procedures.

Response:

Thank you for your comment. In response to your feedback, we have revised the discussion section of our paper to clearly discuss the need for standardized surgical procedures as a foundational step for integrating ML technologies in the OR. We recognize that standardization not only enhances the efficiency and quality of surgical workflows but also serves as a crucial enabler for the development and application of predictive models like ours across different clinical settings.

Additionally, we acknowledge the importance of developing a cross-clinic data platform to enhance ML systems' accuracy and scalability. We're planning to apply our model to new datasets and contribute further to this research area. We appreciate your valuable feedback and look forward to continuing contributing to innovative solutions in this field.

Change(s):

We provide more details on the need for standardizing surgical procedures to integrate ML technologies in the discussion of our paper and mention the need to develop new data sets, especially across clinics.

Comment 4

The first of the most significant effects of RSNs described in the discussion, the reduction of verbal communication, may also change the efficiency of some parties.

Response:

Thank you for your interesting comment. The reduction of verbal communication in the OR can indeed impact the efficiency of the OR team, affecting various parties in different ways. This can be viewed from several perspectives.

On the one hand, less verbal communication can lead to a quieter environment, allowing surgeons and other OR staff to concentrate better on their tasks. This can potentially reduce errors and improve the precision of surgical procedures. In addition, a potential reduction can encourage the standardization of procedures, as teams rely more on established guidelines and signals, enhancing efficiency by minimizing misunderstandings and the need for verbal instructions.

On the other hand, a reduction in verbal communication might hinder the team's ability to coordinate in critical situations where immediate responses are needed. However, with the establishment of RSNs exploring the optimal balance between verbal and non-verbal communication will be an ongoing research area.

Change(s):

We consider the merging of voice input and predictive capabilities in the discussion section to focus not only on reducing communication in the OR but rather to derive an optimal trade-off depending on the surgical phases.

Comment 5

Time variations due to misidentification and emergencies would have a significant impact on efficiency. The mention of the safety for this system is insufficient, and it would be good to also mention methods for dealing with misidentification, etc.

Response:

Thank you for your valuable feedback on our manuscript. We acknowledge the importance of addressing the potential impacts of delays caused by the RSN, as well as the need for a more comprehensive discussion on the safety of the proposed system. In response to your comments, we have made several amendments to the discussion section of our paper to address these concerns more thoroughly. Specifically, we have highlighted the indispensable role of experienced human scrub nurses in ensuring the safety and efficiency of the surgical process. We now propose a hybrid operational model where a human scrub nurse oversees the RSN in the roll-out phase. This approach not only combines the strengths of human expertise and technological innovation but also establishes a critical safety net during the transition phase to more autonomous systems.

We now further propose the incorporation of voice input capabilities into the RSN system. This feature is designed to enable real-time corrections and adjustments by surgeons or supervising scrub nurses. Moreover, by allowing the RSN to communicate the predicted instruments to the surgeon through voice announcements before the actual handover, the system provides an opportunity for immediate correction of any mispredictions, thereby preventing delays and ensuring the continuity of the surgical process. These updates aim to address your concerns by outlining a more robust and safe approach to integrating RSN technology into surgical workflows.

Change(s):

We outline potential measures to improve safety during deployment in our discussion section. For example, we suggest that a human scrub nurse will oversee the RSN in the transition phase to step in if needed in safety-critical phases of surgical procedures. Also, we invite future research to merge voice and predictive capabilities to allow surgeons to correct wrong predictions verbally but also for the

RSN to communicate predictions before handover to reduce the risk for delays. All changes are highlighted in the attached document.

Comment 6

As for Equations 3 through 6, they are common sense statistical formulas and should be removed unless there is something special about them. Or in supplemental material section.

Response:

Thank you for the suggestion to move Equation 3 to 6 to the supplementary material. However, they differ from the standard formulas typically found in ML studies. We consider two different time windows here, namely prediction and resting window. It depends on the timing of the prediction whether a predicted instrument can count as a true positive (TP) or a false positive (FP). The same applies to the missing prediction, which, depending on the timing, is classified as a true negative (TN) or a false negative (FN). We have expanded the standard formulas for Precision, Recall, Accuracy, and the F1 Score accordingly, to include the subdivision into error categories depending on the time window, and see an urgent need to arrange the equations under Figure 5. This ensures the reader's understanding.

Change(s):

Since the equations differ due to the distinction between resting and prediction windows and we have extended the standard formulas, these are essential for understanding the evaluation of the system and therefore remain in the paper in the explanation of the evaluation metrics.

REVIEWERS' COMMENTS:

Reviewer #1 (Remarks to the Author):

Thank you very much for the opportunity to re review

I am satisfied with the authors' responses.

Reviewer #2 (Remarks to the Author):

This paper aims to predict scrub nurse the instrumentation using machine learning to control a scrub nurse robot. Although the accuracy of the prediction would not be sufficient, the paper discusses how more accurate prediction can be expected by adding information on additional information, and we believe that this paper is significant, including future developments.

Peer review comments are appropriately addressed and corrected in the revised version. About comment 6 of me, I understand your comments and agree with your comment.